# DiskHIVF: Disk-Resident Hierarchical Inverted File Index For Billion-scale Approximate Nearest Neighbor Search

## Abstract

The in-memory algorithms for approximate nearest neighbor search (ANNS) has demonstrated remarkable success. However, as the scale of vector data grows, the memory demands of in-memory indexing become increasingly prohibitive. A promising solution lies in hybrid memory-disk implementations, which offload the bulk of data storage to cost-efficient devices such as Solid State Drives (SSDs) while retaining only frequently accessed data in memory. Despite this, existing hybrid memory-disk indexing methods suffer from memory overheads that scale proportionally with the number and dimensionality of the vectors, limiting their memory savings to a modest 5–20×. In this paper, we introduce the Disk-Resident Hierarchical Inverted File Index (DiskHIVF), a novel hybrid memory-disk indexing algorithm with a memory space complexity of $O(\sqrt{N} \cdot d + N)$, where $N$ is the number of vectors and $d$ is their dimensionality. Leveraging its superior space complexity, DiskHIVF achieves several hundred times memory savings compared to the original vectors, and 10–30× reduction compared to state-of-the-art methods. Experimental results on four different datasets demonstrate that DiskHIVF is 1.2-2.3× faster than the state-of-the-art hybrid indexing solutions at achieving the same recall quality of 90%. These results indicate that our approach can significantly reduce the overhead of machine resources while maintaining high search performance.

## 1 Introduction

Approximate Nearest Neighbor Search (ANNS) is fundamental to information retrieval, driving applications like search engines (Huang et al., 2020; Grbovic & Cheng, 2018), recommendation systems (Covington et al., 2016; Okura et al., 2017), and retrieval-augmented generation (RAG) for large language models (LLMs) (OpenAI, 2023; Asai et al., 2023). By identifying vectors closest to a query vector in high-dimensional space, ANNS enables efficient matching of relevant items. Numerous algorithms (Bentley, 1975; Friedman et al., 1977; Sproull, 1991; Beis & Lowe, 1997; Liu et al., 2004; Muja & Lowe, 2014; Wang & Li, 2012; Wang et al., 2012; 2018; Baranchuk et al., 2018) have been designed to improve this process, aiming for high recall and low latency, which are crucial for real-time and large-scale data tasks.

The rapid adoption of LLMs has dramatically increased the volume and dimensionality of vector data, with modern benchmarks like the Massive Text Embedding Benchmark (MTEB) (Muennighoff et al., 2022) highlighting vectors reaching up to 8192 dimensions. This explosion in data has made in-memory indexing prohibitively expensive, spurring interest in hybrid memory-disk solutions that offload the bulk of data storage to cost-efficient SSDs while retaining only a small subset in memory (Chen et al., 2021; Subramanya et al., 2019; Ren et al., 2020; Wang et al., 2024; Ni et al., 2023).

Existing hybrid memory-disk ANNS methods can be broadly divided into navigation-graph-based and inverted-file-based approaches. Among navigation-graph-based solutions, DiskANN (Subramanya et al., 2019) have been widely adopted in industrial applications due to their efficiency and scalability. DiskANN compresses vectors in memory using Product Quantization (PQ) (Jégou et al., 2010) while storing full-precision vectors on disk. During queries, it traverses a navigation graph and refines search results by loading full-precision vectors from disk. However, frequent disk I/O

operations introduce significant latency, limiting its efficiency. To mitigate the high cost of disk I/O, Starling (Wang et al., 2024), a successor to DiskANN , was proposed. Starling constructs a sampled navigation graph in memory to quickly identify vertices close to the query vector as starting points, thereby shortening the search path. Moreover, it introduces a block shuffling algorithm to reorganize the graph index layout on disk, ensuring that vertices and their neighbors are stored within the same disk page. A disk page, typically 4096 bytes, is the smallest unit of I/O operations. By enhancing data locality within each disk page, Starling effectively reduces the number of disk accesses required during search operations. Despite its advantages, our experiments reveal that Starling faces critical challenges when dealing with high-dimensional datasets, such as GIST (Dat, n.d.). Specifically, as vector dimensionality increases, the size of individual vertices often exceeds the 4096-byte limit of a disk page, leading to runtime errors. This constraint significantly undermines Starling's ability to handle high-dimensional vectors and adapt to the rapid growth in vector dimensionality seen in modern applications. These limitations highlight the need for more scalable and robust solutions for high-dimensional ANNS tasks.

Inverted-file-based methods, such as SPANN (Chen et al., 2021), offer another approach to memory-disk ANNS. SPANN uses a hierarchical balanced clustering algorithm to partition the dataset and constructs a memory-resident spatial partition tree (SPTAG) (Chen et al., 2018) for centroids. The member vectors within each cluster are stored on SSDs. However, SPANN requires retaining approximately 16% of vectors as centroids in memory, and each vector may be duplicated up to $8 \times$ on disk. These characteristics result in significant memory and disk overhead. As outlined above, while existing methods achieve some reduction in memory usage, their memory overhead still scales proportionally with the number of vectors ($N$) and their dimensionality ($d$), limiting their efficiency in large-scale scenarios.

In this paper, we propose DiskHIVF, a disk-resident hierarchical inverted file approach, building on the inverted file principles of methods like SPANN and GNOIMI (Chen et al., 2021; Babenko & Lempitsky, 2016). As illustrated in Figure 1, DiskHIVF utilizes a two-level k-means clustering algorithm to partition the data space into $n \times m$ cells, where $n$ and $m$ represent the number of cluster centers at the first and second levels, respectively. Vectors in the dataset are assigned to $n \times m$ cells, which are stored on disk. In memory, only $n + m$ cluster centers and $n \times m$ pointers to the corresponding inverted lists on disk are retained. Typically, the values of $n$ and $m$ are chosen to be approximately $\sqrt{N}$, resulting in a memory space complexity of $O(\sqrt{N} \cdot d + N)$, where $N$ is the number of vectors and $d$ is their dimensionality. To further optimize query efficiency, we propose a centroid reordering algorithm to reorganize the layout of cells on disk. This algorithm ensures that cells frequently accessed together during searches are stored in adjacent disk locations, thereby improving I/O efficiency. Additionally, we developed a query-aware dynamic pruning algorithm that allocates different search cell budget numbers for different queries to reduce the number of I/O operations. Through these designs, DiskHIVF significantly reduces memory overhead while maintaining high retrieval accuracy and optimizing query efficiency. This approach is particularly well-suited for modern large-scale vector search tasks, providing excellent scalability and robustness. Our specific contributions are summarized as follows:

- We propose DiskHIVF, a hybrid memory-disk ANNS algorithm with a memory complexity of $O(\sqrt{N} \cdot d + N)$, achieving 10–30$\times$ memory savings compared to state-of-the-art methods. This means that only a few gigabytes of memory are needed to provide retrieval services for billion-scale vectors.

- Experimental results demonstrate that DiskHIVF is 1.2-2.3$\times$ faster than the state-of-the-art hybrid indexing solutions at achieving the same recall quality of 90% while significantly reducing memory costs across four different datasets.

Furthermore, we will open source our code to facilitate future research and industrial applications.

## 2 RELATED WORK

Given a dataset $P \in \mathbb{R}^{N \times d}$, consisting of $N$ vectors, each characterized by d-dimensional features, and a query vector $p \in \mathbb{R}^d$, the objective of vector search is to identify the nearest neighbor $p^* \in P$. This nearest neighbor is formally defined as $p^* = \arg\min_{q \in P} \text{Dist}(p, q)$, where $\text{Dist}(p, q)$ denotes a chosen distance metric.

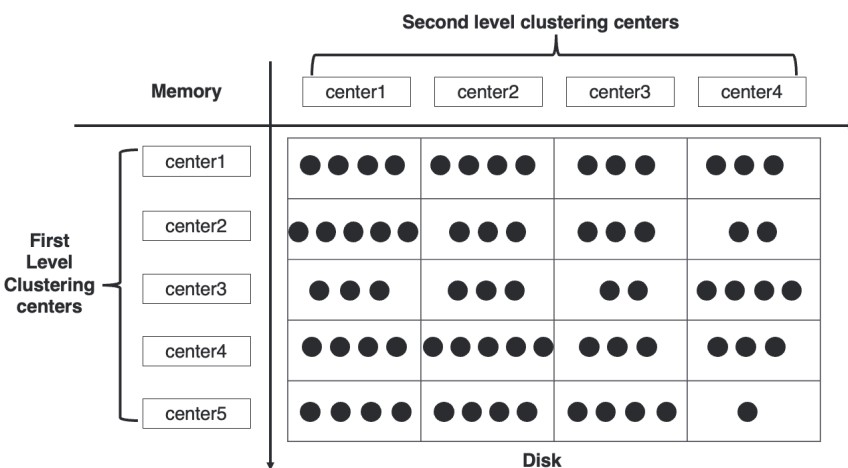

Figure 1: DiskHIVF layout. The first- and second-level centers are stored in memory. Black dots represent dataset vectors assigned to cells stored on disk.

Similarly, the concept of $K$-Nearest Neighbors (K-NN) extends vector search to identify the $K$ closest vectors to a given query vector $p$. Since exhaustive search is computationally prohibitive and results in high query latency, Approximate Nearest Neighbor Search (ANNS) algorithms have been developed to efficiently retrieve approximate $K$-nearest neighbors from large-scale datasets within acceptable time limits. Existing ANNS methods predominantly focus on achieving fast, high-recall searches in memory. These include approaches based on hashing (Datar et al., 2004; Xu et al., 2011), trees (Bern, 2015; Wang et al., 2014), graphs (Malkov & Yashunin, 2018; Dong et al., 2011; Fu et al., 2019), and hybrid techniques (Chen et al., 2018; Iwasaki & Miyazaki, 2018).

However, the exponential growth in vector dataset sizes has rendered memory a critical bottleneck for supporting large-scale vector search. To mitigate memory constraints, one popular strategy is to leverage Product Quantization (PQ) (Jégou et al., 2010) in-memory solutions, such as FAISS-PQ (Johnson et al., 2019), IVFADC (Jégou et al., 2011), and IVFOADC+G+P (Baranchuk et al., 2018). These methods apply PQ to compress vectors and store them in memory efficiently. For instance, the Inverted Multi-Index (IMI) (Babenko & Lempitsky, 2014) partitions the feature space into multiple orthogonal subspaces, creating separate codebooks for each subspace. This enables the construction of a Cartesian product of the feature space across these subspaces. GNO-IMI (Babenko & Lempitsky, 2016) further improves upon IMI by utilizing non-orthogonal codebooks to generate more effective centroids. Although such methods can reduce memory usage to under 64GB for datasets containing 1 billion 128-dimensional vectors, they often experience a significant trade-off in accuracy due to the lossy nature of vector compression. For instance, IMI and similar approaches typically achieve a recall@1 rate of approximately 60%.

Alternatively, hybrid memory-disk systems like DiskANN (Subramanya et al., 2019), Starling (Wang et al., 2024), and SPANN (Chen et al., 2021) combine compressed in-memory indices with full-precision vectors on disk. DiskANN and Starling perform graph search using compressed distances, then re-rank with disk-resident full-precision vectors. SPANN uses a clustering-based approach, storing centroids in memory and keeping disk pointers to member vectors.

Hybrid memory-disk ANNS techniques effectively balance recall quality and retrieval latency, making them a preferred solution for billion-scale vector indexing. Their ability to optimize both memory usage and query performance positions them as a critical enabler of large-scale nearest neighbor search in the era of rapidly expanding vector datasets. However, their memory overhead still scales proportionally with the number of vectors ($N$) and their dimensionality ($d$), limiting their efficiency in large-scale scenarios. Building on recent advancements in hybrid memory-disk ANNS research, we introduce DiskHIVF, a cutting-edge indexing algorithm designed to further optimize resource efficiency. DiskHIVF with a memory complexity of $O(\sqrt{N} \cdot d + N)$ achieves significant reductions in memory usage while maintaining superior retrieval speeds, offering an advanced solution for large-scale vector search applications.

## 3   DISKHIVF

### 3.1   HIERARCHICAL CLUSTERING

Consider a dataset $P = \{p_1, \ldots, p_N\}$ consisting of $N$ points in d-dimensional space. To partition this space efficiently, we train two codebooks, $S = \{S_1, \ldots, S_n\}$ and $T = \{T_1, \ldots, T_m\}$, using a hierarchical k-means algorithm. The codebooks $S$ and $T$ contain $n$ and $m$ codewords, respectively, with $n, m \ll N$. Each codeword is a d-dimensional vector, i.e., $S, T \subseteq \mathbb{R}^d$. The hierarchical clustering divides the data space into $n \times m$ cells, and the points in the dataset are assigned to these cells. The center of each cell is defined as $c_i^j = S_i + T_j$, where $i = 1, \ldots, n$ and $j = 1, \ldots, m$. Thus, the region of the data space corresponding to each cell is defined as:

$$C_i^j = \left\{ x \in \mathbb{R}^d \ \middle| \ i, j = \arg\min_{k,l} \|x - (S_k + T_l)\|^2 \right\} \tag{1}$$

where $C_i^j$ denotes the cell corresponding to the indices $i$ and $j$. Each point $x$ is assigned to the cell $C_i^j$ that minimizes the distance between $x$ and the center $S_i + T_j$.

We employ the hierarchical k-means (HKM) algorithm (Hartigan & Wong, 1979; Babenko & Lempitsky, 2016) to construct the cell centers as described in Equation (1). Since the codebooks $S$ and $T$ we need to train have only a limited number of parameters, we don't need to use all the original vectors for training. In practice, we've found that using 10% of the original vectors as the training set suffices (see Appendix B.1).

To improve training efficiency, we sample 10% of the points from the original vectors as the training set $P'$. As shown in Algorithm 1, we initialize the matrix $D \in \mathbb{R}^{N \times d}$ with the training set $P'$. During each iteration, we first perform k-means clustering (Hartigan & Wong, 1979) on $D$ to obtain $n$ first-level cluster centers $S \in \mathbb{R}^{n \times d}$. Subsequently, we compute the residuals by subtracting the nearest first-level centers from the vectors in the training set $P'$.

Then, we perform a second-level k-means clustering on these residual vectors to obtain $m$ second-level cluster centers $T \in \mathbb{R}^{m \times d}$. Finally, we update the matrix $D$ with the residuals between the vectors in the training set $P'$ and their nearest second-level centers, and this updated $D$ will participate in the next iteration.

After multiple iterations, the algorithm converges and produces two compact codebooks, $S$ and $T$. Unlike SPANN's hierarchical clustering algorithm (Chen et al., 2021), which generates a large number of cluster centers (approximately 16% of the dataset size), our approach leverages shared second-level centers across all first-level clusters, requiring only $n + m$ cluster centers in total. This design significantly reduces the memory overhead while maintaining clustering effectiveness.

---

**Algorithm 1** Hierarchical K-means Clustering

**Require:** Training set $P'$ with $N'$ $d$-dimensional vectors, number of epochs $epoch\_num$, number of first-level cluster centers $n$, number of second-level cluster centers $m$

**Ensure:** first-level cluster centers $S$, second-level cluster centers $T$

1: Initialize $D \leftarrow P'$
2: **for** $epoch = 1$ to $epoch\_num$ **do**
3:     Perform first-level k-means clustering on $D$ to obtain $n$ cluster centers $S$
4:     **for** each vector $p_i$ in $P'$ **do**
5:         Find the nearest first-level cluster center $s_j$ in $S$
6:         Compute the residual $r_i \leftarrow p_i - s_j$
7:         Update $D[i] \leftarrow r_i$
8:     **end for**
9:     Perform second-level k-means clustering on $D$ to obtain $m$ cluster centers $T$
10:     **for** each vector $p_i$ in $P'$ **do**
11:         Find the nearest second-level cluster center $t_k$ in $T$
12:         Compute the residual $r_i \leftarrow p_i - t_k$
13:         Update $D[i] \leftarrow r_i$
14:     **end for**
15: **end for**
16: **Return:** $S, T$

---

### 3.2   INDEX BUILDING

In this subsection, we describe the process of assigning the points in the dataset $P$ to different cells based on the codebooks $S$ and $T$. For each vector $p \in P$, the nearest center indices $i$ and $j$ are determined by solving the following optimization problem:

$$i, j = \arg\min_{k,l} \|p - (S_k + T_l)\|^2 \tag{2}$$

A naive computation of this objective requires evaluating all $n \times m$ combinations for each vector $p$, resulting in a computational cost of $O(nmd)$. This complexity becomes prohibitive for large-scale datasets. To address this issue, we employ a heuristic pruning strategy. By expanding the squared Euclidean distance, the objective function can be reformulated as:

$$\|p - (S_k + T_l)\|^2 = \|p - S_k\|^2 - 2pT_l + (T_l^2 + 2S_k T_l) \tag{3}$$

From Equation 3, $\|p - S_k\|^2$ represents the squared distance between the query point $p$ and the first-level cluster center $S_k$. To optimize computational efficiency, we restrict the evaluation to the top $r$ first-level cluster centers closest to $p$, where $r \ll n$. This pruning reduces the solution space, effectively narrowing the number of candidate evaluations from $n \times m$ to $r \times m$.

Another optimization involves the global precomputation of the term $(T_l^2 + 2S_k T_l)$. For each point $p$, we compute $\|p - S_k\|^2$ and $pT_l$ only once, resulting in a complexity of $O((n + m)d)$. Using these precomputed terms, each candidate solution can subsequently be evaluated in $O(1)$ operations. Consequently, evaluating $r \times m$ solutions requires only $O(rm)$ operations. After computing the $r \times m$ solutions, the optimal cell for the point $p$ is determined, and $p$ is assigned accordingly. The overall time complexity for this optimized process is $O((n + m)d + rm)$, which is a significant improvement over the naive approach's complexity of $O(nmd)$.

Once the assignment process is complete, the disk addresses of the $n \times m$ cells are recorded using pointers. In addition, as outlined in § 3.1, memory is required to store the $n + m$ cluster centers. Typically, the values of $n + m$ are selected such that $n, m \approx \sqrt{N}$. Under these conditions, the overall space complexity of the method is approximately $O(\sqrt{N} \cdot d + N)$. Compared to the space complexity of DiskANN (Subramanya et al., 2019) and Starling (Wang et al., 2024), which is $O(N \cdot d')$, where $d'$ refers to the dimensionality of the product-quantized vectors (typically $1/4$ to $1/32$ of the original dimensionality $d$), and the space complexity of SPANN (Chen et al., 2021), which is $O(C \cdot d)$, where $C$ represents the number of centroids stored in memory (typically $16\%N$), our method has significant advantages in processing large-scale datasets.

## 3.3 DISK LAYOUT

Similar to existing IVF-based methods (Almalawi et al., 2016; Pan et al., 2020; Wang, 2011; Kim et al., 2022), cells that are spatially proximate are more likely to be accessed concurrently during searches. Storing these spatially adjacent cells in consecutive disk locations optimizes read operations and improves disk cache utilization.

To achieve this, we introduce a simple yet effective centroid reordering algorithm that independently reorganizes the first-level and second-level cluster centers to enhance spatial locality. The details are outlined in Algorithm 2.

Using this algorithm, spatially proximate cluster centers are assigned adjacent positions, and their indices are reordered accordingly. Based on the updated indices of the first-level and second-level centers, we perform disk layout reordering. The corresponding cells are reordered with priority given to the first-level center indices, followed by the second-level center indices. The reordered cells are then written to disk sequentially. This reorganization of the disk layout ensures that cells in close spatial proximity are stored in contiguous disk locations, thereby maximizing the efficiency of disk reads and improving overall system performance.

---

**Algorithm 2** Centroid Reordering Algorithm

**Require:** $C = \{v_1, v_2, \ldots, v_n\}$, where $v_i \in \mathbb{R}^d$
**Ensure:** A new sequence of centers $R = \{r_1, r_2, \ldots, r_n\}$
1: Initialize $R = \emptyset$
2: Find $v_{\min} \in C$ such that $\|v_{\min}\| \leq \|v_i\|$ for all $v_i \in C$
3: Add $v_{\min}$ to $R$
4: Remove $v_{\min}$ from $C$
5: $r_{\text{prev}} \leftarrow v_{\min}$
6: **while** $C \neq \emptyset$ **do**
7:     Find $v_{\text{nearest}} \in C$ such that $\|v_{\text{nearest}} - r_{\text{prev}}\| \leq \|v_i - r_{\text{prev}}\|$ for all $v_i \in C$
8:     Add $v_{\text{nearest}}$ to $R$
9:     Remove $v_{\text{nearest}}$ from $C$
10:     $r_{\text{prev}} \leftarrow v_{\text{nearest}}$
11: **end while**
12: **return** $R$

---

## 3.4 ONLINE SEARCHING

This subsection describes the process of retrieving $K$ nearest neighbors for a query vector $q$. Let $r$ denote the number of first-level centers to search and $L$ denote the number of cells to search. The

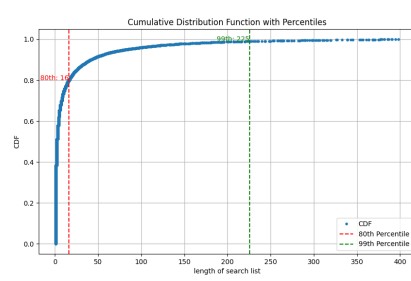

(a) Cumulative Distribution of Searched Cells.

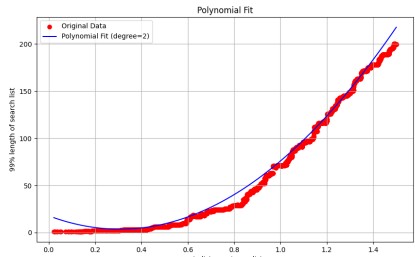

(b) Polynomial Fitting for Query Search Budgets.

Figure 2: Visualization on the SIFT1M Dataset.

search process is divided into three main steps: (1) **Candidate Cell Selection:** Similar to the process of finding the nearest cells during index building, the query process identifies $L$ candidate cells. The candidate cell generation proceeds as follows: Compute the distances between $q$ and all first-level centers in $S$, and select the $r$ nearest first-level cluster centers, denoted as $\{S_{k_1}, \ldots, S_{k_r}\}$. For each selected first-level center, compute the distances between $q$ and the corresponding $r \times m$ cells: cellsDistance$[k_i, l] = \|q - S_{k_i}T_l\|^2 - 2qT_l + (T_l^2 + 2S_{k_i}T_l), k_i \in \{k_1, \ldots, k_r\}, l \in \{1, \ldots, m\}$. Based on cellsDistance, select the $L$ cells closest to $q$, denoted as searchCells. (2) **Disk Access Optimization:** Using searchCells, fetch the inverted lists from disk to obtain the candidate vector set, denoted as candidateVecs. A strategy of merging consecutive read operations is employed here: if multiple cells in searchCells are stored consecutively on disk, they are fetched in a single disk read operation. This optimization, enabled by the prior disk layout reordering (see §3.3), significantly reduces the number of disk reads. We verify the effectiveness of this method in our ablation studies. (3) **Final Distance Computation and Sorting:** Finally, compute the distances between $q$ and all vectors in candidateVecs. Sort these distances and return the top $K$ nearest neighbors as the result.

### 3.5 QUERY-AWARE DYNAMIC PRUNING

In SPANN (Chen et al., 2021), it is noted that different queries may require varying numbers of centers to be searched. Similarly, as shown in Figure 2a, on the SIFT1M dataset, we observe that to retrieve the top-1 result, 80% of queries only require searching 16 cells, while 99% of queries necessitate searching up to 225 cells. Furthermore, as depicted in Figure 2b, we observe that the smaller the distance to the current locally optimal solution, the fewer additional cells need to be searched. In an extreme case, if the distance to the locally optimal solution is zero, no further cells need to be searched.

Based on this observation, we propose a simple query-aware dynamic pruning method. This method uses the distance to the current locally optimal solution to predict the number of cells (denoted as budget$_L$) that need to be searched. Once the number of searched cells exceeds the predicted budget$_L$, the search process is terminated. The overall formulation is as follows:

$$x = \frac{\text{currentDist}}{\text{avgDist}} \tag{4}$$

where *currentDist* represents the distance to the currently identified locally optimal solution, and *avgDist* is the average distance of all points in the dataset to their nearest indexed center.

The query budget$_L$, is modeled as a quadratic polynomial function of $x$:

$$\text{budget}_L = f(x) + \delta \tag{5}$$

where $f(x)$ is a quadratic polynomial defined as $f(x) = ax^2 + bx + c$. The coefficients $a$, $b$, and $c$ are determined through least-squares fitting using the 99% query coverage curve (refer to Figure 2b). This data-driven approach ensures that the polynomial accurately models the distribution of required query budgets, minimizing manual parameter tuning. Additionally, a positive hyperparameter $\delta > 0$ is introduced to vertically shift the polynomial, ensuring that the predicted query budget is sufficiently conservative to accommodate the vast majority of cases. This adjustment mitigates underestimation risks while maintaining computational efficiency.

Table 1: The statistics of experimental datasets (DF: Distance function).

| Dataset | Datatype | Dimensions | Base data | DF | Query data | Query type |
|---|---|---|---|---|---|---|
| SIFT1M (Dat, n.d.) | float | 128 | 1M | l2 | 10k queries | ANNS |
| GIST (Dat, n.d.) | float | 960 | 1M | l2 | 1k queries | ANNS |
| BIGANN (Dat, n.d.) | uint8 | 128 | 1B | l2 | 10k queries | ANNS |
| DEEP (Babenko & Lempitsky, 2016) | float | 96 | 1B | l2 | 100k queries | ANNS |

## 4 EXPERIMENTS

### 4.1 DATASET

As highlighted in Starling (Wang et al., 2024), segmenting large-scale data into multiple parts and allocating these segments to resource-constrained containers (e.g., 2 GB memory, 10 GB disk) is a critical scenario for balancing index construction time and service performance (Guo et al., 2022; Wang et al., 2021). To comprehensively evaluate the performance and resource consumption of our proposed method across various scenarios, we employed four publicly available real-world datasets. These datasets vary in scale, dimensionality, and domain. Table 1 summarizes the details.

### 4.2 IMPLEMENTATION DETAILS

To ensure fair comparisons with related studies, all methods were implemented in C++ on machines with identical configurations. The instance used for index building is configured with 32 CPUs, 256 GB of RAM, and a 4 TB SSD. Meanwhile, the instance used for search tasks has a maximum memory limit of 32 GB. For each competing method, we utilized its recommended configurations . The reported results represent the averages of three runs under identical configurations. In DiskHIVF, we used the same default hyperparameter configuration method across all four datasets to simplify the configuration. The number of primary cluster centers $n$ and secondary cluster centers $m$ were typically set to $n = \sqrt{N/10} \times 2.5$ and $m = \sqrt{N/10} \div 2.5$, where $N$ represents the size of the dataset. Additionally, the number of primary centers considered during the index-building process, $r_{\text{building}}$ (see §3.2), was uniformly set to 200 . Across all datasets, we sample 10% of the original vectors as the training set. The k-means algorithm (see §3.1) is executed for 20 iterations, hierarchical clustering is conducted for 5 iterations, and the parameter $\delta$ (see §3.5) is uniformly set to 10. For a detailed analysis of hyperparameter settings, please refer to the Appendix B.

### 4.3 COMPARISON METRICS

The following metrics were used to evaluate performance: (1) **Recall:** Recall@R measures the proportion of the top $R$ vector identifiers (vector IDs) returned by the ANNS that match the ground truth vector IDs. Given that multiple data vectors may share the same distance to the query vector, we account for this by substituting some ground truth vector IDs with those of vectors that share the same distance to the query during recall calculation. (2) **Latency:** Latency refers to the average query response time, measured in milliseconds (ms). (3) **Memory Index Overhead:** We define the portion of the index that different methods need to load into memory during search operations as the memory index overhead, which is measured in megabytes (MB).

### 4.4 MAIN RESULTS

Due to the inability of competing methods to function reliably or efficiently under memory constraints comparable to our method, we evaluate the search performance of our approach against existing methods using their recommended configurations. Additionally, we report the memory overhead for each method to demonstrate the superiority of our approach.

#### 4.4.1 MEMORY OVERHEAD

Table 2 presents the latency and index memory overhead of our method when recall@1 reaches 90%. Due to the limitation of Starling by the disk page size (4096 bytes), it cannot support the high-dimensional vector dataset, gist (960 dimensions), and thus its metrics are not shown in the results for the gist dataset. Additionally, SPANN exceeded our machine's memory limits when building indexes for the 1B-scale DEEP dataset, as noted in SPFresh, where SPANN requires over 260GB

Table 2: Comparison of latency (Lat, ms) and index memory overhead (Mem, MB), under the condition that Recall@1 reaches 90%.

| Method | SIFT1M | | GIST | | BIGANN | | DEEP | |
|---|---|---|---|---|---|---|---|---|
| | Lat | Mem | Lat | Mem | Lat | Mem | Lat | Mem |
| DiskANN | 1.435 | 50 | 16.273 | 205 | 7.938 | 32768 | 11.938 | 32675 |
| SPANN | 0.964 | 181 | 30.006 | 1207 | 6.042 | 32524 | \ | \ |
| Starling | 1.697 | 30 | \ | \ | 5.902 | 32425 | 36.538 | 32625 |
| DiskHIVF | **0.422** | **2.9** | **10.058** | **6.9** | **4.910** | **1213** | **6.291** | **1210** |
| Gains | 2.3× | 10× | 1.6× | 30× | 1.2× | 27× | 1.9× | 27× |

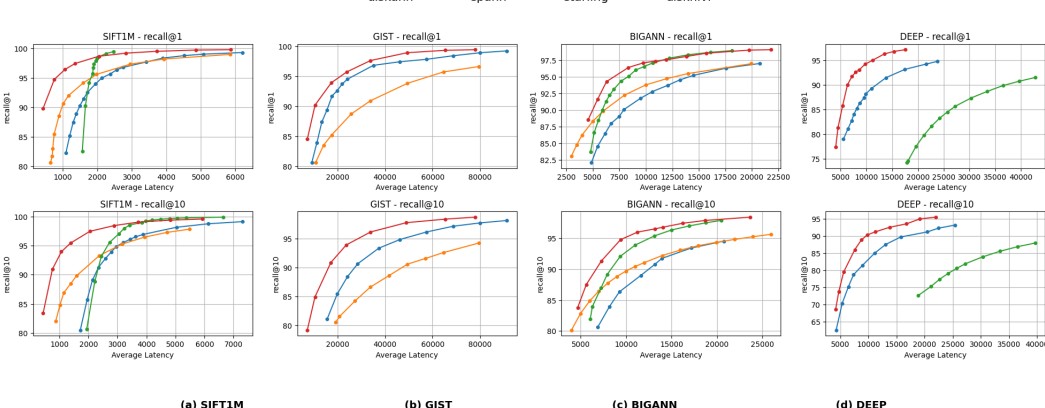

Figure 3: Performance Comparison Across four Datasets.

of memory to build a 1B-level index. Therefore, SPANN's metrics are not shown in the results for DEEP. As shown in Table 2, when recall@1 reaches 90%, our method shows a latency 1.2 to 2.3 times higher than existing methods. Moreover, the memory overhead of our method is significantly lower than that of existing methods. Compared to Starling, which currently performs the best in low-dimensional vector retrieval, our memory overhead is reduced by 10 to 27 times. Compared to the widely used DiskANN, our memory overhead can be reduced by 17 to 30 times. Compared to SPANN, our method reduces memory usage by 27 to 172 times. This means that our method can significantly reduce machine resource consumption while maintaining superior performance.

### 4.4.2 SEARCH PERFORMANCE

In this subsection, we will report in detail the recall accuracy of our method under different query budgets. Figure 3 provides a detailed display of the search performance curves of our method and existing approaches across four datasets of varying scales. As shown in Figure 3(a), our method significantly outperforms existing approaches on the SIFT1M dataset. When recall@10 reaches 95%, the search speed of our method is 2.1 times faster than that of Starling, the state-of-the-art method. The results in Figure 3(b) demonstrate that on the GIST dataset, our method substantially outperforms two strong baseline methods, DiskANN and SPANN, in terms of recall@10. At a recall@10 of 95%, our retrieval speed is 1.6 times faster than that of DiskANN. The results on the billion-scale BIGANN dataset with int8 data type are presented in Figure 3(c). Our method surpasses existing approaches in recall@10, achieving a performance 1.4 times faster than Starling at 95% recall@10 accuracy. Figure 3(d) shows the experimental results on the billion-scale DEEP dataset with float data type. Due to the larger scale of this dataset compared to BIGANN, DiskANN and Starling, which rely on product quantization, are affected to varying degrees under the 32GB memory constraint. In contrast, our method benefits from superior memory space complexity and continues to perform excellently under such memory limitations. Using only 1.2GB of memory, our method achieves a performance 1.6 times faster than DiskANN at 90% recall@10 accuracy. These experiments demonstrate that our method exhibits advantages in retrieval speed compared to existing approaches.

Table 3: Disk Access Count Under Different Search Cell Numbers $L$.

| Method | 400 | 900 | 1600 | 2500 | 3600 | 5000 |
|---|---|---|---|---|---|---|
| DiskHIVF | 151.633 | 282.392 | 438.781 | 612.477 | 797.77 | 1000.94 |
| w/o Merge-Read | 323.228 | 708.617 | 1239.67 | 1914.99 | 2733.88 | 3769.18 |

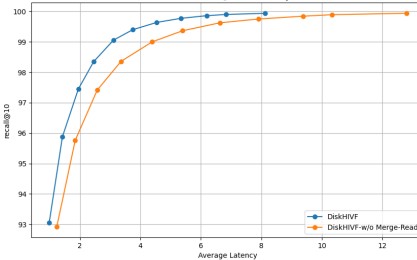

Figure 4: Effectiveness of the Disk Access Optimization Strategy.

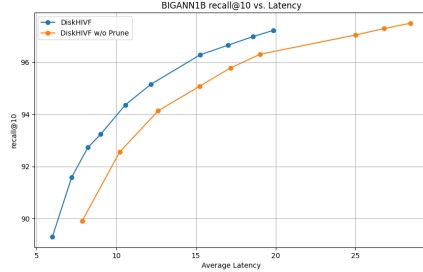

Figure 5: Effectiveness of Query-Aware Dynamic Pruning.

## 4.5 ABLATION STUDIES

In this subsection, we conduct ablation studies on disk access optimization and query-aware dynamic pruning in our proposed method.

**Effect of disk access optimization**. To validate the effectiveness of the disk access optimization strategy proposed in §3.4, we conducted comparative experiments using the SIFT1M dataset. From Table 3, it is evident that during the search process, employing the strategy of merging consecutive read operations significantly reduces the number of disk accesses. Figure 4 further illustrates the impact of this strategy on retrieval performance. The results demonstrate that the recall rates under various query latency budgets are consistently better when the strategy is applied, compared to when it is not. This confirms that the merged consecutive read strategy effectively improves retrieval performance by optimizing disk access patterns, and consequently versifying the effectiveness of our centroid reordering algorithm and disk layout reordering (§3.3).

**Effect of query-aware dynamic pruning**. As introduced in §3.5, to efficiently handle diverse queries during online search, we incorporated query-aware dynamic pruning. This technique reduces the number of cells that need to be searched by pruning unnecessary entries from the candidate list. Figure 5 compares the performance on the BIGANN dataset with and without query-aware dynamic pruning. The results indicate that query-aware dynamic pruning can further reduce query latency without compromising recall. Notably, this technique not only decreases query latency but also reduces resource usage during querying.

## 5 CONCLUSIONS AND FEATURE WORK

In this paper, we introduce DiskHIVF, a disk-memory hybrid algorithm for approximate nearest neighbor search that aims to achieve extremely low memory complexity. DiskHIVF achieves memory savings of several hundred times while attaining state-of-the-art performance in terms of recall and query latency. Unlike existing disk-memory hybrid methods that require a trade-off between memory overhead and search speed, DiskHIVF employs a disk-resident hierarchical inverted index architecture. This innovative design significantly reduces the memory required to store centroids while retaining fine-grained spatial partitioning. Consequently, DiskHIVF achieves dual advantages: substantial memory reduction under low-latency constraints and high recall. Extensive experimental evaluations confirm the advantages of DiskHIVF. Compared to existing disk-memory hybrid methods, DiskHIVF reduces memory usage by tens to hundreds of times while establishing new state-of-the-art benchmarks in latency and recall. In future work, we aim to extend the DiskHIVF framework to support incremental vector insertions and further optimize the query-aware dynamic pruning algorithm to enhance performance.

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

## A LIMITATIONS

### A.1 LIMITATIONS OF DISTANCE FUNCTIONS

The method proposed in the paper relies on specific properties of the L2 distance function, as shown in Equation 3, and currently only supports cosine similarity, L2 (Euclidean distance), and IP (Inner Product). When two vectors are normalized, the L2 distance equals 2 - 2 * cos. IP2COS (Morozov & Babenko, 2018) is a conversion method that transforms IP distance into cosine distance. The distance values in the search results are always L2 distances. It does not support other types of distance

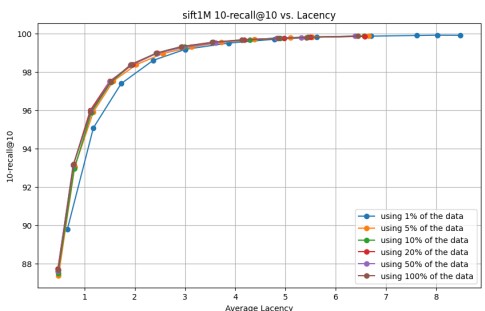

Figure 6: Impact of Different Training Data Sampling Ratios.

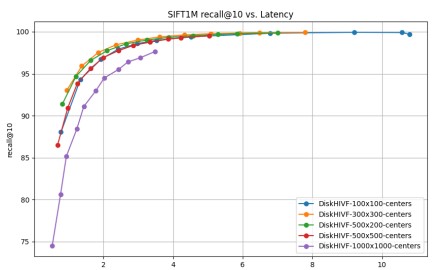

Figure 7: Impact of Different Numbers of Primary and Secondary Centroids.

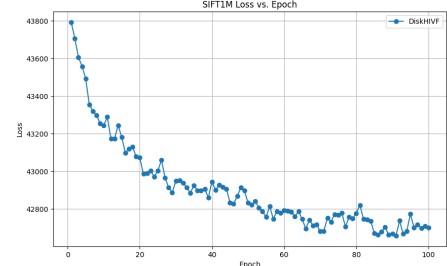

Figure 8: Number of Iterations for Hierarchical Clustering.

functions (such as the L1 distance function). Currently, most application scenarios primarily use L2 distance (Huang et al., 2020; Grbovic & Cheng, 2018; Covington et al., 2016; Okura et al., 2017; Asai et al., 2023), and scenarios requiring L1 or other distance metrics are relatively few. Therefore, our current research is mainly focused on L2 distance. We will follow up with support for other distances in future research.

### A.2 DEPENDENCE ON CLUSTERING QUALITY

Like most ANNS algorithms based on inverted indices (Johnson et al., 2019; Babenko & Lempitsky, 2014; Chen et al., 2021; Babenko & Lempitsky, 2016), our method also relies on the quality of k-means clustering. Under extreme data distributions, the uneven clustering problem in k-means clustering can lead to the formation of large clusters, which may negatively impact retrieval performance. However, thanks to our hierarchical index structure, our method can generate a large number of cells with very few cluster centers, resulting in an average of only about 10 vectors per cell under default parameter settings. Taking the DEEP dataset as an example, in extreme cases, the largest cell may contain about 9,000 vectors, far exceeding the average, but this is still an acceptable range. Additionally, searching through large cells often leads to finding better local optimal solutions quickly, triggering our dynamic pruning strategy(see §3.5) that reduces the number of cells searched for that query. Moreover, we have designed a caching strategy to store frequently accessed large cells in memory, further mitigating the impact of large cells on query performance. Regarding the other extreme, the case of empty cells, we simply skip them during the candidate cell search phase.

## B PARAMETER SENSITIVITY

### B.1 DIFFERENT SAMPLING RATIOS OF TRAINING DATA

Figure 6 illustrates the impact of using data with different sampling ratios as the training set on retrieval performance. It can be observed that when the sampling ratio exceeds 10%, the improvement

in retrieval performance becomes minimal. This supports our inference that, because the hierarchical clustering algorithm requires fewer parameters to be trained, it does not need the full dataset for training.

## B.2 DIFFERENT NUMBERS OF CENTROIDS

In this subsection, we conduct a set of experiments to study the impact of the number of first-level centroids and second-level centroids on the retrieval performance. Figure 7 shows the recall@10 and latency on the SIFT1M dataset when different numbers of first-level centroids and second-level centroids are selected. It can be observed that better results are obtained when the product of the number of first-level centroids $n$ and the number of second-level centroids $m$ is close to 10% of the number of points $N$ in the dataset. This can be understood as that when the average number of vectors assigned to each single cloud cell space is about 10, a better balance can be achieved between the quality of space partitioning and the disk reading speed. At the same time, as described in §3.4, the time complexity of querying the candidate cell list is $O((n + m)d + rm)$, which can also be written as $O(dn + (d + r)m)$. It can be seen that the coefficient of $n$ in the time complexity is smaller than that of $m$. Therefore, $m$ is typically set to be less than or equal to $n$. Under the constraint that $n \times m = N/10$, we adjusted the value of $n$ to be 1 to 8 times that of $m$, and found that when $n = \sqrt{N/10} \times 2.5$ and $m = \sqrt{N/10} \div 2.5$, satisfactory performance can be achieved in most cases.

## B.3 DIFFERENT EPOCHS OF HIERARCHICAL CLUSTER

In this subsection, we prove through experiments the convergence of the hierarchical clustering algorithm described in §3.1. Figure 8 shows the loss values of the SIFT1M dataset under different numbers of iterations. Here, the loss value refers to the average value of the shortest distances from all points in the dataset to the index centroids. It can be seen that as the number of iterations increases, the loss value shows an obvious convergence trend. However, in practical applications, we found that when the number of iterations exceeds 5, although the loss value is still decreasing, the improvement in recall quality is already very small. Therefore, in order to save training time, it is usually sufficient to set the number of iterations of hierarchical clustering to 5.

