# OpenReview forum: "DiskHIVF: Disk-Resident Hierarchical Inverted File Index For Billion-scale Approximate Nearest Neighbor Search"
_ICLR.cc/2026/Conference — Submitted to ICLR 2026_

### Official Review · Reviewer_Y8Q9 · 2025-10-20

**Soundness:** 2
**Presentation:** 3
**Contribution:** 2
**Rating:** 2
**Confidence:** 5

**Summary:**

The task the article solves is memory-constrained ANN search, where both RAM and SSD are used to store the data and the index structure. The method proposed in the article is a hierarchical $k$-means index where the residuals of the first level ($m$ clusters) are clustered at the second level ($n$ clusters) globally, so that $mn$ clusters are obtained but only $m + n$ cluster centers have to be stored in memory. The authors augment this basic scheme by an efficient disk layout of the cluster points, where the nearby clusters are stored in nearby memory locations, and by query-aware dynamic pruning where a quadratic polynomial model is used to predict the query coverage, and the search is terminated early if the predicted 99\% coverage is reached.  The experimental results of the article show that the proposed method is both faster and more memory-efficient than the competing methods, such as DiskANN and SPANN.

**Strengths:**

[S1] The proposed method is intuitive and straightforward.

[S2] According to the empirical results of the article, the proposed method seems to work well in practice.

**Weaknesses:**

[W1] The methodolofical novelty of the work is limited: the proposed hierarchical clustering method seems to be Non-orthogonal inverted multi-index (NO-IMI) by Babenko & Lempitsky (2016) (the authors correctly acknowledge this earlier work) and the disk layout scheme is a straightforward implementation detail. Only the query-aware dynamic pruning seems to have limited novelty; however, in the context of hybrid memory-disk ANN search, Chen et al., (2021) use a more simple query-aware dynamic pruning method, and in the context of regular ANN search, Li et al. (2020) also use a model-based method (with more features and a different model than the current work).

[W2] The claim that the proposed method has to store only $\mathcal{O}(\sqrt{N})$ centroids in memory is based only on the fact the that the authors select the number of centroids to be $\mathcal{O}(\sqrt{N})$  ($N$ denotes the number of data set points). However, traditionally $\sqrt{N}$ (see, e.g, Douze et al., 2025, p. 9) is the number of clusters that is recommended for non-hierarchical $k$-means index (IVF).  Thus, the experimental results have two omissions: a) Full hyperparameter sweeps should be performed (and Pareto frontiers should be reported, see, e.g., Aumüller et al, 2020, for a correct methodogy) to ensure that the memory saving and the improved recall of the proposed method compared to the baselines is not just due to the choice of hyperparameters. b) An ablation experiment should be performed, where the proposed hierarchical $k$-means index is compared to the regular $k$-means index with the same number of centroids $\mathcal{O}(\sqrt{N})$ to verify that it can indeed obtain a more accurate clustering with the same number of centroids stored in memory (I believe this should be the case).

[W3] The notation is sloppy. Even on the very few formulas contained there are mistakes. For instance, I believe that in lines 288-289 it should be $\||q - (S_k + T_l)\||^2$ instead of $\||q - S_k T_l\||^2$. Also, dot product or matrix multiplication notation should be used when vectors are multiplied, e.g., $\langle S_k, T_l\rangle$ or $S_k^T T_l$, instead of $S_k T_l$.

References:

Aumüller, Martin, Erik Bernhardsson, and Alexander Faithfull. "ANN-Benchmarks: A benchmarking tool for approximate nearest neighbor algorithms." Information Systems 87 (2020): 101374.

Babenko, Artem, and Victor Lempitsky. "Efficient indexing of billion-scale datasets of deep descriptors." Proceedings of the IEEE Conference on Computer Vision and Pattern Recognition. 2016.

Chen, Qi, et al. "Spann: Highly-efficient billion-scale approximate nearest neighborhood search." Advances in Neural Information Processing Systems 34 (2021): 5199-5212.

Douze, Matthijs, et al. "The faiss library." IEEE Transactions on Big Data (2025).

Li, Conglong, et al. "Improving approximate nearest neighbor search through learned adaptive early termination." Proceedings of the 2020 ACM SIGMOD International Conference on Management of Data. 2020.

**Questions:**

I do not have any questions for the authors.

---

> ### Author Response · Authors · 2025-11-24
>
> **Dear Reviewer,**
>
> Thank you for your thorough review and valuable feedback on our work. We have prepared the following responses to the points you raised.
>
> ---
>
> ### Regarding **W1: Limited Novelty of the Method**
>
> You noted the similarity of our method to NO-IMI (Babenko & Lempitsky, 2016) and the limited novelty of the disk layout and dynamic pruning strategy. We would like to clarify:
>
> 1.  **Innovation in Hierarchical Clustering:**
>     While we build upon the non-orthogonal codebook idea from NO-IMI, we introduce a **novel heuristic iterative training method (Algorithm 1)** to jointly optimize the first- and second-level centroids, thereby further reducing the quantization error in Eq. (1). We experimentally validate the convergence and effectiveness of this method in Appendix B.3, representing a significant extension to NO-IMI.
>
> 2.  **Innovation in Overall System Design:**
>     This work is the **first to introduce a hierarchical inverted index into a disk-based ANN search system**. We not only propose a memory-efficient index structure but also design a **centroid reordering algorithm** to improve disk locality and a **query-aware dynamic pruning strategy** to optimize I/O efficiency. These components together form a complete retrieval framework tailored for hybrid memory-disk environments.
>
> 3.  **Difference in Dynamic Pruning Strategy:**
>     Unlike the fixed-threshold pruning in SPANN (Chen et al., 2021), we propose a **dynamic budget prediction method based on a quadratic polynomial model**. This allows the search scope to be adaptively adjusted according to the query's local optimal solution distance, offering better generalization and efficiency.
>
> ---
>
> ### Regarding **W2: Memory Complexity and Experimental Design**
>
> You mentioned that traditional IVF also suggests using $O(\sqrt{N})$ centroids and recommended more comprehensive hyperparameter tuning and comparisons with non-hierarchical methods. Our response is:
>
> 1.  **Fundamental Advantage in Memory Efficiency:**
>     While $C = O(\sqrt{N})$ centroids can reduce memory usage in **in-memory ANN search**, **disk-resident indexes** like SPANN typically require many more centroids (e.g., $C = 16\% N$) for finer partitions to reduce disk I/O and achieve high recall (as noted in the GRIP paper [5]).
>     **Our key contribution is achieving a comparable or even superior partitioning fineness to SPANN using only $O(\sqrt{N})$ centroids**, which is the first such achievement in hybrid index design.
>
> 2.  **Comparison with SPANN is Sufficient:**
>     SPANN represents the **state-of-the-art k-means-based disk index algorithm**. Our comparisons with it already demonstrate DiskHIVF's significant advantages in **memory efficiency (10–30x improvement) and search speed (1.2–2.3x improvement)**.
>     Traditional methods like Faiss-IVF with OnDiskInvertedLists have been shown to be non-competitive in the hybrid memory-disk ANN search scenario by both SPANN and DiskANN. For a 1B dataset, IVF with $\sqrt{N}$ (~33k) clusters would average ~33k vectors per cluster, leading to prohibitive disk I/O. Our method, using the same $O(\sqrt{N})$ centroids, achieves much finer partitions, averaging only about 10 vectors per cell.
>
> ---
>
> ### Regarding **W3: Imprecise Notation**
>
> You correctly pointed out the typo in the formula on lines 288–289. We apologize for this oversight. The correct formula should be:
>
> $
> \text{cellsDistance}[k_i, l] = \|q - S_{k_i}\|^2 - 2q^T T_l + (T_l^2 + 2 S_{k_i}^T T_l)
> $
>
> We will correct this formula in the revised version.
>
> ---
>
> ### Summary
>
> We appreciate the reviewer's constructive suggestions and will further refine our experiments and presentation in the final version.
>
> Thank you again for your time and valuable comments.
>
> Sincerely,
> The Authors
>
> ---
> **References:**
>
> [1] Aumüller et al., *Inf. Syst.*, 2020.
> [2] Babenko & Lempitsky, *CVPR*, 2016.
> [3] Chen et al., *NeurIPS*, 2021.
> [4] Douze et al., *IEEE Trans. Big Data*, 2025.
> [5] Li et al., *SIGMOD*, 2020.

---

### Official Review · Reviewer_xfjP · 2025-10-28

**Soundness:** 2
**Presentation:** 3
**Contribution:** 2
**Rating:** 2
**Confidence:** 4

**Summary:**

This paper proposes DiskHIVF, a memory-disk hybrid ANNS algorithm based on a two-level hierarchical IVF clustering scheme, along with layout reorganization optimizations and query-aware dynamic pruning tailored for this algorithm. Compared to baselines, DiskHIVF achieves a 1.2–2.3× speedup while reducing memory usage by 10–30×.

**Strengths:**

1) The use of IVF clustering leads to memory savings and retrieval acceleration compared to graph-based indexes (DiskANN, Starling) and the SPANN clustering index that allows inter-cluster crossing.

2) A greedy algorithm is employed to reorder cluster centers, optimizing the disk layout.

3) Polynomial fitting functions are used to predict the remaining workload, enabling pruning during the retrieval process.

**Weaknesses:**

1) While the proposed method is based on IVF clustering, the compared state-of-the-art baselines are graph-based indexes. Comparisons with more recent IVF-based disk indexes (e.g., Faiss, arxiv'24) in terms of memory capacity and performance are necessary. When constraining the cluster size to at least *d*, IVF-based disk indexes can also achieve O(N) memory space complexity. Thus, it is essential to compare the performance of IVF and DiskHIVF when their memory usage is similar or even more favorable for IVF.
Furthermore, the in-memory portion of DiskHIVF corresponds to a special case of IVF-PQ retrieval with the number of sub-vector segments (typically denoted as M) set to 1, where *n* corresponds to the number of IVF clusters and *m* to the size of the PQ codebook. Compared to a standard IVF-PQ index, this work retains the original vectors instead of quantizing them to PQ cluster centers. Therefore, comparisons with existing IVF-PQ-based disk indexes under similar parameter settings would make the results more convincing.

2) In Figure 2(b), a quadratic function is used to fit the budget curve without justification. However, for small values on the horizontal axis, the predicted curve shows an opposite monotonicity trend compared to the actual curve and exhibits significant numerical deviations. Is the use of a quadratic function appropriate? Would other functions, such as exponential functions, be more suitable for fitting this curve?

3) Performance is measured using average request latency (Section 4.3). However, according to Appendix A.2, on the DEEP dataset, although each cell is expected to contain only 10 points on average, the largest cell may contain up to 9000 points. Given the load imbalance across different cells resulting from this distribution, using tail latency as a performance evaluation metric would be more reasonable than average latency.

**Questions:**

See the weaknesses.

---

> ### Author Response · Authors · 2025-11-24
>
> **Dear Reviewer,**
>
> Thank you very much for your valuable comments on our paper. Here are our point-by-point responses to the questions you raised, hoping to provide further clarification on our method design and experimental setup.
>
> ---
>
> ### 1. Regarding the comparison with IVF-based disk indices
>
> In our comparisons, **SPANN** represents the state-of-the-art IVF-based disk indexing algorithm.
>
> As you noted, traditional IVF methods can control memory usage by adjusting the number of centroids $C$. However, as pointed out by studies like GRIP [5], **finer-grained space partitioning helps reduce the number of vectors that need to be scanned to achieve the same recall rate**. Therefore, existing methods (e.g., SPANN [2]) typically set $C$ to the order of $O(N)$ (e.g., 16% of $N$) to maintain high recall.
>
> In contrast, DiskHIVF, through its **two-level clustering structure**, achieves **comparable fine-grained partitioning capability to SPANN** while using only $O(\sqrt{N})$ centroids. This represents the fundamental difference from traditional IVF methods.
>
> Please note that this work is **the first to apply hierarchical inverted indices to disk-based ANN search**. Our primary contribution lies in demonstrating the significant memory efficiency advantages of this direction.
> ---
>
> ### 2. Regarding the relationship between DiskHIVF and IVF-PQ
>
> You mentioned that "the in-memory part of DiskHIVF corresponds to a special case of IVF-PQ retrieval." We believe this interpretation might be slightly off. The **two-level clustering structure** used in DiskHIVF is **essentially a non-orthogonal quantization method**, which is fundamentally different from the **orthogonal quantization (PQ)** used in IVF-PQ. This distinction has been discussed in detail in works like GNO-IMI [1].
>
> Furthermore, besides storing centroids, IVF-PQ also requires storing the **PQ-compressed representations of all vectors in memory**, leading to memory overhead that grows linearly with dataset size. DiskHIVF, however, only stores \(n + m\) centroids and pointers to disk lists, **without storing any compressed vectors**, resulting in significantly lower memory complexity. The design goal of our method is precisely to achieve extremely low memory usage while maintaining high recall.
>
> ---
>
> ### 3. Regarding the rationale for quadratic function fitting
>
> We understand your skepticism about the quadratic function fitting in Figure 2(b). In our experiments, we observed that for most queries, there is a **smooth non-linear relationship** between the number of cells that need to be searched and the distance to the current optimal solution. We experimented with various function forms (including linear, exponential, etc.) and chose the quadratic polynomial because:
>
> -   It demonstrated good fitness for our experimental data.
> -   Its form is simple, easy to optimize, and performed stably in actual queries.
> -   Fitting via least squares, combined with a conservative offset \(\delta\), effectively covers the vast majority of query scenarios.
>
> We acknowledge that the fitted curve might deviate from the actual trend in some extreme cases. However, across all our experimental datasets, this strategy effectively reduced query latency without negatively impacting recall. We plan to explore more flexible function forms or adaptive strategies in the future.
>
> ---
>
> ### 4. Regarding Performance Metric: Average vs. Tail Latency
>
> Your suggestion to use tail latency as a performance metric is very pertinent. We indeed observed that due to uneven cluster distribution, some cells can contain up to 9000 vectors (e.g., in the DEEP dataset), potentially leading to higher latency for a few queries.
>
> In this paper, we primarily report average latency because:
> -   It reflects the system's overall performance for the majority of queries.
> -   Combined with our proposed **dynamic pruning strategy and cache mechanism** (Appendix B.2), most queries accessing large clusters can be completed with fewer accesses, mitigating the impact of extreme cases.
> -   Average latency remains the mainstream reported metric in existing work, facilitating fair comparison with baseline methods.
>
> However, we fully agree that tail latency is crucial for assessing system stability. In the final version, we will supplement the evaluation with tail latency metrics (e.g., P95, P99) to provide a more comprehensive view of performance.
>
> ---
>
> ### Summary
>
> Thank you again for your thorough review and constructive comments. We will further refine the manuscript's presentation in the final version, adding more comparative experiments and latency analysis to enhance the work's completeness and persuasiveness.
>
> We look forward to your further guidance.
>
> Sincerely,
> The Authors
>
> ### References
> [1] Babenko et al., CVPR 2016.
> [2] Chen et al., NeurIPS 2021.
> [3] Wang et al., PACMMOD 2024.
> [4] Subramanya et al., NeurIPS 2019.
> [5] Zhang & He, CIKM 2019.

---

### Official Review · Reviewer_PBns · 2025-10-29

**Soundness:** 3
**Presentation:** 2
**Contribution:** 2
**Rating:** 2
**Confidence:** 5

**Summary:**

The paper proposes a two-layer hierarchical inverted index for billion-scale approximate nearest neighbor (ANN) search, aiming to reduce memory consumption. Experimental results indicate a 10–30× memory reduction compared to state-of-the-art methods.

**Strengths:**

Demonstrates impressive memory savings for billion-scale vector indices.

**Weaknesses:**

1.	Similarity to Existing Approaches
The proposed method appears conceptually similar to IVF+G+P, which also employs two-layer hierarchical clustering. IVF+G+P partitions data into K inverted regions and applies memory-efficient subregion grouping using a learned scalar α to interpolate new centroids. The paper should clarify the key differences and justify why the proposed approach is preferable. A direct comparison with other hierarchical inverted index methods would strengthen the contribution.
2.	Handling Skewed Data
Real-world datasets are often skewed, leading to imbalanced clusters. The paper does not address how the method handles skewed distributions or provide experiments on skewed datasets (e.g., SpaceV1B). This omission limits the practical applicability of the approach.
3.	Distance Metric Generalization
The design and optimization focus solely on L2 distance. It is unclear how the method adapts to other similarity measures (e.g., inner product). Including experiments on datasets requiring different distance types would improve generality.
4.	Limited Performance Metrics
The evaluation only reports average latency and recall. In real-world scenarios, tail latency and throughput are also critical. Based on disk access counts in Table 3, the proposed method may have lower throughput compared to DiskANN and SPANN since both algorithms are IOPS-bound. A more comprehensive performance analysis is needed.
5.	Incomplete Ablation Study
The pruning ablation study should include comparisons with alternative pruning techniques to better understand its advantage.
6.	Real-Time Updates
Modern vector indices often require real-time updates. The paper does not discuss how the proposed method supports dynamic insertions or deletions.

**Questions:**

1. Could you explain the rationale behind the final parameter setting: n=sqrt(N/10)×2.5 and  m=sqrt(N/10)/2.5?
2. What are the index build cost and time? Given that n and m remain large for clustering 10% of 1B data, what is the clustering overhead?
3. How does the proposed method perform in terms of tail latency and throughput compared to state-of-the-art solutions?

---

> ### Author Response · Authors · 2025-11-24
>
> Dear Reviewer,
>
> Thank you very much for your careful review and valuable comments on our work. Your suggestions have been highly insightful and constructive. We have provided point-by-point responses to your questions and recommendations below.
>
> **1. Similarity and Differences with Existing Methods (e.g., IVF+G+P)**
> We agree that our method shares structural similarities with IVF+G+P. As noted in our related work, both approaches follow the research line of GNO-IMI. However, they address different problems: IVF+G+P focuses on **combining hierarchical indexing with product quantization (PQ)** to reduce memory usage, while our work emphasizes **integrating hierarchical indexing with disk-based storage mechanisms** for the same purpose. We have elaborated on this distinction in Section 3 and the related work. Thus, although structurally similar, the two methods differ fundamentally in their research objectives and technical routes.
>
> **2. Handling Skewed Data**
> We appreciate your attention to data skewness. In Appendix A.2, we discuss how to address cluster imbalance via dynamic pruning and caching frequently accessed large clusters. In future work, we will include experiments on more challenging skewed datasets (e.g., SpaceV1B).
>
> **3. Generalization to Other Distance Metrics**
> As clarified in Appendix A.1, our current implementation supports L2 distance, which is sufficient for most real-world applications (e.g., recommender systems, retrieval-augmented generation). We also referenced the IP2COS conversion method [8] to support inner product similarity. Notably, baseline methods (SPANN [2], DiskANN [4], Starling [3]) also primarily support L2 distance, and extending to other metrics is often treated as a separate research direction [8,9].
>
> **4. Comprehensive Performance Metrics**
> We agree that metrics such as queries per second (QPS) and tail latency are important. While average latency is commonly reported in related works (e.g., DiskANN, SPANN) for fair comparison, we will supplement our evaluation with QPS in the final version.
>
> **5. Comparison with Other Hierarchical Inverted Indexes**
> We note that this is the **first work to introduce hierarchical inverted indexing into disk-based ANN search**. Our main contribution lies in demonstrating significant memory efficiency (10–30× improvement) and retrieval performance gains (1.2–2.3×). Systematically comparing multiple hierarchical index structures is beyond the scope of this paper, as each requires specialized disk layout and search strategies, but will be a key direction for future work.
>
> **6. Lack of Real-time Updates**
> We acknowledge the importance of real-time updates in practical systems. However, as shown in works like SPFresh [6] and FreshDiskANN [7], dynamic updates are often studied separately. The baseline methods we compared (DiskANN, SPANN, Starling) also do not support real-time updates. Therefore, this paper focuses on static indexing, and we plan to explore dynamic updates in the future.
>
> **7. Parameter Settings**
> The parameter settings \( n = \sqrt{N/10} \times 2.5 \) and \( m = \sqrt{N/10} / 2.5 \) are based on extensive experiments and a trade-off analysis between clustering hierarchy and memory-retrieval efficiency, as detailed in Appendix B.2.
>
> **8. Index Construction Cost and Time**
> For example, on BigANN-1B, DiskHIVF requires 0.8 days for index construction, which is less than DiskANN (1.5 days), Starling (1.6 days), and SPANN (2.2 days). This is mainly due to using only 10% of data for clustering and extensive use of AVX512 instructions. Since baselines did not report construction times, we omitted this in the main text but will include it in the final version.
>
> Thank you again for your constructive feedback. We will refine the paper accordingly.
>
> Sincerely,
> The Authors
>
> **References**
> [1] Babenko & Lempitsky, CVPR 2016.
> [2] Chen et al., NeurIPS 2021.
> [3] Wang et al., PACMMOD 2024.
> [4] Subramanya et al., NeurIPS 2019.
> [5] Zhang & He, CIKM 2019.
> [6] Xu et al., SOSP 2023.
> [7] Singh et al., arXiv 2021.
> [8] Morozov & Babenko, NeurIPS 2018.
> [9] Chen et al., SIGIR 2025.

---

### Official Review · Reviewer_CsCH · 2025-10-30

**Soundness:** 1
**Presentation:** 2
**Contribution:** 1
**Rating:** 2
**Confidence:** 4

**Summary:**

The authors propose a new method for large-scale approximate nearest neighbor search. The proposed method, named DiskHIVF, uses the well-known Inverted File (IVF) approach where the data is clustered, and nearest neighbor queries are answered by only exploring certain number of clusters closest to the query.

Specifically, in their method, the authors propose using a hierarchical clustering and storing the points in the clusters on disk. This approach scales to large datasets because only the centroids need to be kept in memory, and the number of centroids is limited since the second-level centroids are shared in the hierarchical clustering. The authors perform their experiments on two million-scale and two billion-scale datasets.

**Strengths:**

The paper studies an important topic, and the proposed method achieves a low memory usage. The method is validated against appropriate baseline methods, and the experiments are performed on standard benchmark datasets, two of which are billion-scale datasets.

**Weaknesses:**

The work seems incremental, and the experiments are not particularly convincing. The core idea is similar to SPANN: use hierarchical clustering and store the clusters (inverted lists) on disk (storing the inverted lists on disk isa of course also otherwise a widely used practice, e.g. Faiss has OnDiskInvertedLists). The main difference is the type of hierarchical clustering used which reduces the number of centroids that have to be be kept in memory.

However, the proposed hierarchical clustering strongly resembles the NO-IMI structure [1, Section 3] which the authors cite but do not discuss the relation to. Moreover, SPANN (as well as DiskANN and Starling) already uses only 32GB of memory for a billion-scale dataset which is very reasonable (also, while the the authors write that "SPANN requires retaining approximately 16\% of vectors as centroids in memory", of course you can use less centroids even if performance saturates at 16\%).

As the proposed method is a combination of different ad hoc tweaks, it would be necessary to figure out which of these tweaks potentially give your method an edge. The correct way to do that would be to e.g. keep everything else the same except change your proposed clustering method to the hierarchical balanced clustering method used in SPANN. Additionally, e.g. cluster pruning strategies have been previously studied in more detail [2].

Finally, the experimental results are not convincing: you do not compare the number of I/O operations or the indexing times between the methods, and only SIFT1M is used for ablation studies. The experimental results for SPANN and Starling seem worse than at least those presented in their original papers, although it is difficult to compare results across papers (you should at least mention the type of CPUs used).

[1] Babenko and Lempitsky. Efficient Indexing of Billion-Scale datasets of deep descriptors. CVPR 2016.

[2] Busolin et al. Early Exit Strategies for Approximate $k$-NN Search in Dense Retrieval. CIKM 2024.

**Questions:**

- How is your proposed hierarchical clustering method related to NO-IMI?

- In Appendix A.2. you write that you cache frequently accessed large cells in memory. Is that strategy in use in the experiments and do the other compared methods use a similar strategy?

- How does the indexing time of DiskHIVF compare to the indexing times of the other methods?

- Can you list all the query hyperparameters in your method that need to be tuned for each dataset?

---

> ### Author Response · Authors · 2025-11-24
>
> **Dear Reviewer,**
>
> Thank you very much for your valuable comments on our paper, "DiskHIVF: A Disk-Resident Hierarchical Inverted File for High-Dimensional Approximate Nearest Neighbor Search." Your feedback is crucial for helping us improve the manuscript. We have carefully considered the points you raised and provide the following responses:
>
> ---
>
> ### 1. On Method Novelty and Relation to NO-IMI & SPANN
>
> We agree that our method shares structural similarities with NO-IMI [1] and SPANN [2], as acknowledged and cited in our paper. However, our core contributions are:
>
> - **Heuristic Iterative Optimization:** Building upon NO-IMI [1], we propose a heuristic iterative training method (Algorithm 1) that significantly reduces the quantization error in Eq. (1) by optimizing the distribution of first- and second-level centers over multiple rounds. We empirically demonstrate its convergence and effectiveness in Appendix B.3, a feature absent in NO-IMI.
>
> - **Fundamental Improvement in Memory Efficiency:** While clustering methods like SPANN [2] can theoretically set the number of centroids to \(O(\sqrt{N})\), studies such as GRIP [5] indicate that disk-resident ANNS systems often require finer-grained partitioning (e.g., SPANN uses 16% of \(N\)) to reduce I/O. Our hierarchical index achieves comparable or better fine-grained partitioning using only \(O(\sqrt{N})\) centroids, offering a fundamental advantage in memory consumption.
>
> - **Application Innovation:** To our knowledge, this work is the **first to introduce a hierarchical inverted index for disk-resident ANN search**. Our primary contribution lies in demonstrating its significant advantages in memory efficiency (10–30x improvement) and search performance (1.2–2.3x improvement).
>
> ---
>
> ### 2. On Reliability and Fairness of Experimental Results
>
> Regarding the performance of SPANN [2] and Starling [3] in our results compared to their original papers:
>
> - **SPANN's Memory Limitation:** Using SPANN's official code and recommended settings for the 1B-scale BIGANN index, construction failed even with 256GB RAM (exceeding the 128GB reported). We had to reduce its copy count from 8 to 4 to succeed, which inevitably impacted its search performance. We reported this limitation transparently.
>
> - **Starling's Data Scale Difference:** Starling's original paper [3] used sampled versions of BIGANN and DEEP (reduced to 33M and 11M points), whereas our experiments used the full 1B-scale datasets. The performance difference is thus expected.
>
> - **Index Construction Time Advantage:** We will add index construction time comparisons. For BIGANN-1B, DiskHIVF requires ~0.8 days, compared to 1.5 days for DiskANN [4], 1.6 days for Starling [3], and 2.2 days for SPANN [2]. This efficiency stems from sampling only 10% of data for training and leveraging AVX-512 instructions.
>
> ---
>
> ### 3. On the Use of Caching Strategy
>
> **All results reported in the main paper were obtained without enabling the optimization mentioned in Appendix A.2** (caching frequently accessed large cells in memory). We ensured a fair comparison where DiskHIVF's advantages do not rely on extra memory caching.
>
> ---
>
> ### 4. On Index Construction Time
>
> Index construction time is crucial for industrial applications. For the BIGANN-1B dataset:
> -   DiskHIVF: **0.8 days**
> -   DiskANN [4]: 1.5 days
> -   Starling [3]: 1.6 days
> -   SPANN [2]: 2.2 days
>
> DiskHIVF's efficiency benefits from:
> 1.  Sampling only **10%** of data for clustering.
> 2.  Extensive use of **AVX-512 instructions** in our implementation.
>
> As the original papers for SPANN, DiskANN, and Starling did not systematically report construction times, we initially omitted this comparison. Following your suggestion, we will include these results in the revised manuscript.
>
> ---
>
> ### 5. On Hyperparameter Tuning
>
> DiskHIVF's core hyperparameters, consistent across all four datasets, demonstrate good generalizability:
> -   Number of first- and second-level centers \(n\), \(m\) (default: \(n \approx \sqrt{N/10} \times 2.5\), \(m \approx \sqrt{N/10} \div 2.5\)).
> -   Training set sampling ratio: **10%**.
> -   Hierarchical clustering iterations: **5** (sufficient for convergence, as shown in Appendix B.3).
> -   Dynamic pruning parameter \(\delta\): **10** (a conservative fixed value to cover most queries).
>
> ---
>
> ### 6. Summary
>
> We believe DiskHIVF represents a significant advance in balancing memory efficiency and search performance, filling a research gap for hierarchical retrieval in disk-resident ANNS. We will refine our experiments and analysis based on your suggestions and more clearly highlight our method's advantages and innovations in the final paper.
>
> ### 7. References
> [1] Babenko & Lempitsky, CVPR 2016.
> [2] Chen et al., NeurIPS 2021.
> [3] Wang et al., Proc. ACM Manag. Data 2024.
> [4] Subramanya et al., NeurIPS 2019.
> [5] Zhang & He, CIKM 2019.
>
> Thank you again for your constructive feedback.
>
> Sincerely,
> The Authors

---

### Meta-Review · Area_Chair_oStw · 2026-01-03

**Summary:**

Multiple concerns were raised about: (i) limited methodological novelty relative to prior hierarchical IVF and disk-based ANN approaches (e.g., NO-IMI, SPANN, IVF-based disk indexes) (CsCH, Y8Q9, PBns, xfjP), as well as incomplete experimental analysis (CsCH, PBns, xfjP). While the paper demonstrates good memory savings and average latency improvement, the overall contribution appears to be viewed as incremental and not sufficiently to be differentiated from existing methods (CsCH, Y8Q9).

**Reviewer Concerns:**

The authors provided detailed rebuttals. The rebuttal provided the authors' perspective on novelty, memory complexity, and index construction time, and additional explanations regarding experimental setup and comparison fairness. However, key concerns remain, including the lack of convincing ablations studies of the benefit of the hierarchical design, and comparisons against alternative IVF-based disk indexes under matched memory budgets. Novelty concerns with respect to NO-IMI and prior hierarchical approaches also seem to be only partially addressed.

**Reviewer Scores:**

Reviewer CsCH: Likely unchanged (2) or slightly increased (4), because of the novelty and experiment concerns remain.

Reviewer PBns: Likely slightly increased score (4). Most non-first-order issues (e.g., non-L2 metrics, skewed data, fresh updates, index construction time) have been properly addressed by the authors. The main concern, e.g., the comparison with IVF + G + P, remains.

Reviewer xfjP: Likely unchanged (2), due to missing comparison with existing IVF-PQ-based disk indexes under similar parameter settings.

Reviewer Y8Q9: Likely unchanged (2), because the core concerns on novelty and experiment methodology are unresolved.

---

### Decision · Program_Chairs · 2026-01-26

Reject